# Creation and Validation of the European Portuguese Version of the Systemic Lupus Erythematous Quality of Life Questionnaire

**DOI:** 10.3390/ijerph20020897

**Published:** 2023-01-04

**Authors:** Pedro Lopes Ferreira, Rita Novais Cunha, Carla Macieira, Tomás Fontes, Luís Sousa Inês, Ana Maduro, Ana Martins, Frederico Rajão, Carolina Furtado, Anabela Barcelos

**Affiliations:** 1Centre for Health Studies and Research, University of Coimbra, 3004-512 Coimbra, Portugal; 2Faculty of Economics, University of Coimbra, 3004-512 Coimbra, Portugal; 3Rheumatology Unit, Tâmega e Sousa Hospital Centre, 4564-007 Penafiel, Portugal; 4Rheumatology Department, Santa Maria Hospital, Northern Lisbon Hospital Centre, 1649-035 Lisboa, Portugal; 5Research Unit in Rheumatology, iMM Faculty of Medicine, University of Lisbon, 1649-028 Lisboa, Portugal; 6Rheumatology Department, Coimbra Hospital and University Centre, 3004-561 Coimbra, Portugal; 7Faculty of Health Sciences, Beira Interior University, 6201-001 Covilhã, Portugal; 8Rheumatology Department, São João Hospital University Centre, 4200-319 Porto, Portugal; 9Department of Medicine, Faculty of Medicine, University of Porto, 4200-319 Porto, Portugal; 10Rheumatology Department, Algarve Hospital and University Centre, 8000-386 Faro, Portugal; 11Rheumatology Department, Divino Espírito Santo Hospital, 9500-370 Ponta Delgada, Portugal; 12Rheumatology Department, Baixo Vouga Hospital Centre, 3810-501 Aveiro, Portugal; 13Egas Moniz Health Alliance, University of Aveiro, 3810-193 Aveiro, Portugal; 14Comprehensive Health Research Centre, NOVA National School of Public Health, 1150-082 Lisboa, Portugal

**Keywords:** PRO, outcome assessment, quality of life, systemic lupus erythematous

## Abstract

(1) Background: Patients with systemic lupus erythematous (SLE) experience profound effects on health-related quality of life (HRQoL) that cannot be explained by objective indicators of mortality and morbidity. This study aimed to adapt the SLE Quality of Life (SLEQoL) questionnaire to the European Portuguese population and to assess its reliability and validity for patients with SLE. (2) Methods: Two independent translators translated the original version of the SLEQoL questionnaire into Portuguese. A back-translated version was produced. The Portuguese version of the questionnaire was reviewed and tested for validity and reliability. Cronbach’s alpha and the internal validity index were calculated to verify the internal reliability and validity of the content. Rheumatologists filled out the SLE Disease Activity Score (SLE-DAS) and Systemic Lupus International Collaborating Clinics/American College of Rheumatology Damage Index SLICC/ACR-DI questionnaires. (3) Results: This study involved 180 patients, of which 93.8% were females. The results indicated very high internal consistency reliability (α = 0.949), low correlations between the SLEQoL and the SLE-DAS, a correlation between all SLEQoL dimensions and all SF-36 dimensions (except for “response to treatment” and “self-image”), and good correlation scores with both the EQ-5D-5L index and VAS. (4) Conclusion: The Portuguese version of the SLEQoL questionnaire is valid and reliable for the measurement of HRQoL in SLE patients.

## 1. Introduction

Systemic lupus erythematous (SLE) is a chronic autoimmune disease that affects multiple organs with an unpredictable and variable course [1]. Throughout the evolution of the disease, patients with SLE face physical, psychological and social changes [2]. This disease has profound effects on health-related quality of life (HRQoL) that cannot be fully explained by traditional indicators of mortality and morbidity [3].

Currently, the most important purpose of medical care for patients with SLE is to reduce inequalities and disabilities. Thus, the evaluation of quality of life is an important facet of the management of this disease [3].

In addition to generic measures, such as the 36-Item Short Form Survey (SF-36) [4], three major specific measures have been designed in recent decades to assess HRQoL in SLE patients [2]. These are the Lupus Quality of Life (LupusQoL) questionnaire [5], the Systemic Lupus Erythematosus Quality of Life (SLEQoL) questionnaire [1], and the SLE Quality of Life (L-QoL) questionnaire [6]. The main questionnaire used in this study, the 40 item SLEQoL questionnaire, was developed in 2005 by Leong et al. and designed to measure six major dimensions: physical functioning, activities, symptoms, treatment, mood, and self-image).

This study aimed to adapt the SLEQoL questionnaire to the European Portuguese population and to assess its reliability and validity in patients with SLE.

## 2. Materials and Methods

### 2.1. Linguistic and Cultural Adaptation

After proper authorization by the authors of the original version of the SLEQoL questionnaire, two Portuguese translators independently performed two translations from English to Portuguese. A consensus version developed from these two translations led to a back-translated version, which was compared to the original version. The linguistic adaptation was completed with a clinical review by two experts with experience in clinical trials and cognitive debriefing interviews performed with a group of ten patients with characteristics similar to the study population. Following these steps, we also verified the validity of the content of the Portuguese version of the SLEQoL.

### 2.2. Participants

This study involved patients from five hospital units: (i) Baixo Vouga Hospital Centre, Aveiro; (ii) Santa Maria Hospital and University Centre, Lisbon; (iii) Hospital and University Centre, Coimbra; (iv) São João Hospital and University Centre, Porto; and (v) Divino Espírito Santo Hospital, Ponta Delgada, Azores. The sample consisted of patients diagnosed with SLE fulfilling the 1997 revised American College of Rheumatology (ACR) classification criteria and/or the Systemic Lupus International Collaborating Clinics (SLICC) criteria. Moreover, they were able to understand the study, give informed consent, participate in the study, and answer the questionnaires.

Data collection was performed by rheumatologists from the rheumatology departments of these five hospital units. The questionnaires filled out included, in addition to the Portuguese version of the SLEQoL, the Portuguese versions of the five-level EuroQol-5-dimension (EQ-5D-5L) and SF-36 generic instruments. This information was complemented with the patients’ clinical and sociodemographic data, which were provided by the rheumatologist on the day of the consultation. Clinical data included the Systemic Lupus Erythematosus Disease Activity Score (SLE-DAS) and the Systemic Lupus International Collaborating Clinics/American College of Rheumatology Damage Index (SLICC/ACR-DI) measurement instruments.

### 2.3. Ethical Considerations

Anonymity was guaranteed to all participants. The study was initially approved by the Ethics Commission of Baixo Vouga Hospital Center (number 44-03-2020). After this approval, the other hospitals did not require a formal request for ethics authorization. They simply accepted the former decision. All patients were asked to read and sign an informed consent form after having received information about the study.

### 2.4. Measurement Instruments

#### 2.4.1. SLEQoL

In 2005, Leong et al. [1] developed and validated the SLEQoL to assess HRQoL in patients with SLE. It includes 40 items divided into 6 domains: physical function (six items), activities (nine items), symptoms (eight items), treatment (four items), mood (four items), and self-image (nine items) [1]. Each item has a score ranging from 1 to 7 and a total/domain summary score can be obtained as the sum or the mean of the corresponding responses. Therefore, when summing, the total score ranges from 40 to 280, with higher values corresponding to worse quality of life [1]. The authors concluded that the SLEQoL offers better content validity and responsiveness to change than the SF-36 [1]. It is culturally adapted and validated for Chinese [7], Brazilian Portuguese [8], Thai [9], Arabic [10], and French populations [11].

This questionnaire was built by a team of rheumatology professionals who nominated items they considered to be important determinants of the HRQoL of SLE patients. The 51 initial items were assembled into a questionnaire and given to 100 SLE patients. They were invited to suggest items that were omitted in the questionnaire that were considered important to them, although they did not add any items to the list. Patients were also asked to assess the importance and the frequency of occurrence of each item using a seven-point Likert scale. Factor analysis, Rasch model analysis, and an expert review were used for item reduction, resulting in a new questionnaire with 40 questions. This final questionnaire was applied to a cohort of 275 patients to study its psychometric properties.

#### 2.4.2. EQ-5D-5L

The EQ-5D-5L is a generic preference-based measurement instrument designed to provide HRQoL scores [12]. These scores can easily be converted into quality-adjusted life year (QALY) scores to be integrated into cost–utility economic evaluations. The EQ-5D-5L’s descriptive system includes five dimensions (mobility, self-care, usual activities, pain/discomfort, and anxiety/depression) with five levels each. The second part of the EQ-5D questionnaire comprises a standard 20 cm vertical visual analogic scale (EQ-VAS) calibrated from ”the worst health you can imagine” (scored 0) at its base to ”the best health you can imagine” (scored 100). For this version to be used in health economic evaluation, societal values need to be assigned to the 3125 health states generated by this instrument. The Portuguese version of the EQ-5D-5L [13] was created based on a representative sample of the population with 1045 individuals.

#### 2.4.3. SF-36

The SF-36 is an easy and understandable generic instrument used to assess the perception that individuals have regarding their own health status [4]. It consists of a multidimensional questionnaire with 36 items presented in eight scales or domains: physical functioning (ten items), physical role limitations (four items), pain (two items), general health (five items), vitality (four items), social functioning (two items), emotional role limitations (three items), and mental health (five items), as well as one item on reported health transition. For each domain, it is possible to obtain a score of 0–100. Physical and mental component summary scores can also be calculated. Higher scores mean better health status perceptions. The Portuguese version was also created based on a representative sample of the Portuguese population with 930 individuals [14].

#### 2.4.4. SLE-DAS

The SLE-DAS is a disease activity measure for SLE [15]. An initial study with 520 patients showed that the SLE-DAS has good construct validity for the detection of clinically significant changes in disease activity for SLE patients. Using the Physician Global Assessment as a dependent variable, the authors applied multivariate linear regression with the manifestations of the disease as dependent variables. A weighted sum of all the 17 disease manifestations makes it possible to determine a continuous total score [15], which can also be obtained with a calculator (available at: http://sle-das.eu/) (accessed on 16 October 2022)

#### 2.4.5. SLICC/ACR-DI

The SLICC/ACR-DI [16] was developed by the ACR to quantify the damage (persistent for at least six months) that occurs following the onset of SLE in patients with both active and inactive disease. This index has been shown to have content, face, criterion, and discriminant validity and to correlate with mortality. It is used by clinicians and researchers to evaluate accumulated damage and includes assessments of 12 different organ systems (ocular, neuropsychiatric, renal, pulmonary, cardiovascular, peripheral vascular, gastrointestinal, musculoskeletal, and skin, as well as premature gonadal failure, diabetes, and malignancy).

### 2.5. Reliability and Validity

To test the internal consistency reliability, we used the Cronbach alpha indicator, which can range from 0 to 1, with values above 0.70 being considered ideal [17]. The following hypothesis was formulated:

**H1:** *The Portuguese version of the SLEQoL shows good internal consistency*.

For construct validity, we assessed both structural validity and hypothesis testing [17]. For structural validity, we performed factor analysis based on principal components estimates after testing the sampling adequacy through the Kaiser–Meyer–Olkin (KMO) test and Bartlett’s test of sphericity. A KMO score lower than 0.60 was considered poor, between 0.60 and 0.70 was considered fair, between 0.70 and 0.80 was considered average, between 0.80 and 0.90 was considered good, and higher than 0.90 was considered very good [18]. Bartlett’s sphericity test should have an associated significance lower than 0.001.

For hypothesis testing, we drew several hypotheses for known-groups or subsamples based on sociodemographic (sex, age, marital and employment status, years of education) and clinical (duration of the disease, disease activity, and damage that had occurred in different organ systems) characteristics. Student’s t-test was used for two independent variables and ANOVA for more than two independent variables. To correlate the SLEQoL total index with numerical variables, we used Pearson’s correlation coefficient. Correlations lower than 0.30 were considered weak, between 0.30 and 0.50 were moderate, and higher than 0.50 were considered strong [19]. The following three hypotheses were formulated:

**H2:** 
*Exploratory factor analysis replicates the original structure of the SLEQoL.*


**H3:** *SLEQoL is able to discriminate based on sociodemographic variables*.

**H4:** *SLEQoL is able to discriminate based on clinical variables*.

Finally, criterion validity was tested by comparing scores from the Portuguese version of the SLEQoL with the scores obtained with other measuring instruments, such as the generic health status instrument SF-36 and the generic HRQoL instrument EQ-5D-5L. Disease activity (SLE-DAS) and damaged organ systems (SLICC/ACR-DI) indices were also compared with SLEQoL scores. We expected to demonstrate the similarities and differences between measured concepts. We mainly used Pearson’s correlation to test the significance. The following four hypotheses were formulated:

**H5:** *SLEQoL dimensions are correlated with the SF-36*.

**H6:** *SLEQoL dimensions are correlated with the EQ-5D-5L*.

**H7:** *SLEQoL dimensions are correlated with the SLE-DAS*.

**H8:** *SLEQoL dimensions are correlated with the SLICC/ACR-DI*.

Considering that SF-36 is a generic health status measure, a priori we did not expect to obtain significant correlations with the SLEQoL. On the other hand, some significant correlations were expected with both the EQ-5D-5L index and EQ-VAS.

## 3. Results

### 3.1. Linguistic and Cultural Adaptation

Following the internationally recommended methods, a clinical review was performed by two rheumatologists and very minor changes were proposed. Furthermore, the cognitive debriefing was performed in two meetings, about one hour each, in order to test the understanding of the Portuguese version of the SLEQoL. In both meetings, ten patients (eight women and six patients under 45 years of age) were present. The mean filling-out time was 8.9 ± 2.3 min (mode = 7; max = 14). In general, all participants considered the questionnaire to be clear and of an appropriate length and as having questions that were easy to understand, unambiguous, and without redundancies.

### 3.2. Sample

The sample was composed of 180 SLE patients. However, two patients did not want to provide the data corresponding to sex, age, marital and employment status, and education. Another patient did not provide their age. The sample size met the minimum criteria defined for validations of health measurement instruments [17]. A minimum of 100 subjects is considered a good sample size for validity studies. The socioeconomic and clinical characteristics of our sample are presented in Table 1.

The large majority of patients were female (93.8%), married (60.7%), and older than 40 years of age (60.4%). More than 60% of then were employed and had at least 10 years of formal education. The mean disease duration was 15 ± 8.8 years; the mean disease activity, as measured by the SLE-DAS, was 2.5± 3.6; and the damage accrual, as measured by the SLICC/ACR-DI, was 1.5± 1.1.

The perceptions patients had about health status, obtained from the different dimensions of the SF-36, and about HRQoL, obtained from the EQ-5D-5L, are presented in Table 2.

Regarding the SF-36 scores, we noticed that patients showed the lowest perception of health status in the “general health” dimension (44.5), followed by the ”vitality” (49.1), ”pain” (55.8), and ”mental health” (64.8) dimensions. Very few individuals had severe or extreme problems in the ”mobility” (2.3%), “self-care” (1.2%), “usual activities” (2.8%), ”pain/discomfort” (3.9%), and ”anxiety/depression” (1.7%) dimensions. In summary, the global EQ-5D-5L index was 0.85 and the mean EQ-VAS score was 74.1.

### 3.3. Reliability

The internal consistency reliability was measured using Cronbach’s alpha scores and is presented in Table 3.

Looking at the SLEQoL, the worst scores were in the “self-image” and “mood” dimensions and the best were in the “response to treatment” and “physical function” dimensions. As can also be observed, the total SLEQoL score had an excellent internal consistency (H1). All the dimensions also had good internal consistency, with the “response to treatment” dimension showing a lower alpha (0.685).

### 3.4. Validity

The structural validity was tested using factor analysis. The KMO score was good (0.885) and Bartlett’s test of sphericity was associated with a *p* < 0.001. Similarly to the authors of the original version [1], we found the following six factors explained the 40 items of the SLEQoL: (F1) physical functioning (items 1–6) and physical symptoms (items 21–23); (F2) mood and self-image (items 16–19) and low self-esteem (item 34); (F3) social and occupational activities (items 7–13 and 28–31) and the embarrassment question (item 35); (F4) unpredictability of the response to treatment (items 36–39), including exposure to the sun (item 14) and making less money (item 15); (F5) self-esteem (items 32–35); and (F6) unpleasant aspects of the treatment (items 24–27). The variance explained by these six factors was 65% (H2).

Table 4 presents the results from testing the discriminatory power of the SLEQoL with regard to sociodemographic variables.

As shown in this table, the SLEQoL total scores were shown to be only determined by patient age, number of years of education, and employment status, meaning that individuals older than 54 years old with less years of education or who were not employed had the tendency to report higher scores, meaning lower HRQoL (H3). We found a nonsignificant correlation between SLEQoL total score and disease duration (0.075; *p* = 0.362) (H4).

Finally, to test the criterion validity, the different SLEQoL scores were compared with SF-36 dimensions and EQ-5D-5L scores, as well as with SLE-DAS and SLICC-ACR/DI scores. Table 5 shows the results. Testing H5 and using the above criteria, all SLEQoL dimensions except for the “response to treatment” and “self-image” dimensions were correlated with all SF-36 dimensions. To be more explicit, total SLEQoL score was highly correlated with SF-36 mental dimensions, SLEQoL physical function with physical dimensions, and SLEQoL “symptoms” and “mood” with mental dimensions. Regarding the EQ-5D-5L scores, we found higher correlations between SLEQoL “total score”, “physical function”, “activities”, and “symptoms” and both the EQ-5D-5L index and EQ-VAS (H6). Moreover, a nonsignificant correlation was found when the entire SLEQoL was correlated with the SLE-DAS (0.152; *p* = 0.062) (H7). However, when correlated with SLICC/ACR-DI scores, we found a significant correlation (0.479; *p* = 0.002) (H8).

## 4. Discussion

The obvious advantage of using specific measures to assess HRQoL in SLE patients is that they better capture the symptoms and issues that characterize the disease, naturally increasing their sensitivity to changes over time [2,20].

The Portuguese version of the SLEQoL was written in simple language, like the original version, and showed good psychometric indicators. To validate this Portuguese version, we collected a sample of 180 SLE patients. In general, they were female, older, and had a long disease duration, similarly to other studies [8,10,11].

The internal consistency reliabilities of the entire SLEQoL and its dimensions were good, as also shown for the original version [1] and other versions, such as the Brazilian [9] and Chinese [8] versions. Furthermore, the factorial structure obtained for the Portuguese version was very similar to that obtained by the original authors [1].

The scores from the Portuguese version of the SLEQoL were higher (meaning lower HRQoL) in individuals older than 54 years old who had less education or were unemployed. When correlating the SLEQoL scores with health status, quality of life and damage indicators mainly had strong correlations with SF-36 physical and emotional dimensions. Similar results were found in other languages [8,10]. However, when comparing SLEQoL scores with disease activity and damage, our results were incompatible with other publications [8,9,21,22,23], suggesting that activity and damage may not be the major factors that interfere with quality of life. Other determinants, such as patients’ personalities and coping skills, can influence their perceptions of HRQoL.

SLE predominantly affects women; however, men are affected by a more severe disease [24]. In fact, the limited number of SLE males included in this study did not allow us to reach conclusions about SLE males’ HRQoL in this population, which was a limitation of this study.

A strong point of this study was the participation of patients from the north, center, south, and islands, providing a good representation of the Portuguese population.

Finally, addressing the limitations of this study, we may refer to the lack of a longitudinal aspect aimed at measuring the sensitivity of the SLEQoL to changes in disease activity. For disease duration, SLE-DAS score, and SLICC index especially, we found a wide distribution of data. As a result, the corresponding standard deviations were substantially larger compared to the means. A narrower inclusion criterion and/or a substantially larger sample might have solved this potential limitation. Furthermore, since at the beginning of this study, there were no other specific instruments assessing QoL for SLE patients adapted to European Portuguese, we were not able to compare SLEQoL scores with any other specific measures.

## 5. Conclusions

This study aimed to adapt a questionnaire on HRQoL for patients with SLE to the Portuguese population. The Portuguese version of the SLEQoL was successfully produced and validated. It can be used for evaluating SLE patients in routine clinical practice, as well as in research in Portugal. Other studies are needed to test its sensitivity to change.

## Figures and Tables

**Table 1 ijerph-20-00897-t001:** Sample demographic and clinical characteristics (n = 180).

Variable	Value	No.	%
Sex	Female Male Missing	167 11 2	93.8 6.2
Age (years)	(18–24) (25–39) (40–54) ≥55 Missing Missing Minimum–maximum Mean ± standard deviation	14 56 77 30 3 18–77 43.2 ± 12.6	7.9 31.6 43.5 16.9
Marital status	Single Married Widowed/divorced/separated Missing	48 105 22 2	27.0 60.7 12.3
Employment status	Employed Retired Unemployed Other Missing	116 30 13 19 2	65.2 19.6 7.2 10.6
Years of education	(0–4) (5–9) (10–12) ≥13 Missing	17 53 54 54 2	9.6 29.8 30.3 30.3
Disease duration (years)	Minimum–maximum Mean ± standard deviation	25.0–41.3 14.8 ± 8.8	
Disease activity (SLE-DAS index)	Minimum–maximum Mean ± standard deviation	0.37–32.6 2.5 ± 3.6	
Damage to organ systems (SLICC index)	Minimum–maximum Mean ± standard deviation	1–6 1.5 ± 1.1	

**Table 2 ijerph-20-00897-t002:** Health status and quality of life.

	Variable	Value	No.	%
SF-36	Physical function Physical role limitations Pain General health Vitality Social function Emotional role limitations Mental health	m ± sd m ± sd m ± sd m ± sd m ± sd m ± sd m ± sd m ± sd	71.5 ± 23.6 70.6 ± 26.1 58.8 ± 24.1 44.5 ± 19.4 49.1 ± 24.0 71.2 ± 24.9 71.6 ± 26.4 64.8 ± 23.2	
	Physical summary measure Mental summary measure	m ± sd m ± sd	44.3 ± 9.9 47.5 ± 10.7	
EQ-5D-5L	Mobility	No problem Slight problems Moderate problems Severe problems Extreme problems	121 34 21 3 1	67.2 18.9 11.7 1.7 0.6
	Self-care	No problem Slight problems Moderate problems Severe problems Extreme problems	145 21 12 1 1	80.6 11.7 6.7 0.6 0.6
	Usual activities	No problem Slight problems Moderate problems Severe problems Extreme problems	95 51 28 4 1	53.1 28.5 15.6 2.2 0.6
	Pain/discomfort	No problem Slight problems Moderate problems Severe problems Extreme problems	56 70 47 7 0	31.1 38.9 26.1 3.9 0.0
	Anxiety/depression	No problem Slight problems Moderate problems Severe problems Extreme problems	78 62 37 2 1	43.3 34.4 20.6 1.1 0.6
	PT-Index (0.00–1.00) VAS (1–100)	m ± sd m ± sd	0.85 ± 0.14 74.1 ± 18.3	

m = mean; sd = standard deviation; VAS = visual analogic scale.

**Table 3 ijerph-20-00897-t003:** SLEQoL internal consistency.

Variable	Items	Mean ± sd	Internal Consistency
Physical function Occupational activity Symptoms Response to treatment Mood Self-image Total score	6 9 8 4 4 9 40	1.9 ± 1.0 2.2 ± 1.3 2.3 ± 0.9 1.7 ± 0.9 2.7 ± 1.5 2.4 ± 1.1 2.2 ± 0.9	0.880 0.905 0.780 0.685 0.921 0.857 0.949

m = mean; sd = standard deviation.

**Table 4 ijerph-20-00897-t004:** SLEQoL total scores for different levels of sociodemographic and clinical variables.

Variable	Value	n	Mean	sd	|t| or F	*p*-Value
Sex	Female Male	167 11	2.2 2.3	0.9 0.8	0.315	0.753
Age	(18–25) (25–40) (40–55) ≥55	14 56 77 30	2.2 2.0 2.2 2.8	1.0 0.8 0.8 1.0	6.110	<0.001
Marital status	Married Not married	108 72	2.3 2.1	0.9 1.0	0.915	0.362
Years of education	≤9 ≥10	70 108	2.6 2.0	1.0 0.8	4.107	<0.001
Employment status	Employed Not employed	116 64	2.0 2.6	0.8 1.0	4.099	<0.001

m = mean; sd = standard deviation.

**Table 5 ijerph-20-00897-t005:** Correlations between SLEQoL and SF-36, EQ-5D-5L, SLE-DAS, and SLICC/ACR-DI.

		SLEQoL						
		Total Score	Physical Function	Activities	Symptoms	Treatment	Mood	Self-Image
SF-36	Physical function Physical role limitations Pain General health Vitality Social function Emotional role limitations Mental health	−0.643 −0.714 −0.640 −0.591 −0.740 −0.643 −0.700 −0.735	−0.730 −0.657 −0.588 −0.495 −0.534 −0.503 −0.607 −0.467	−0.650 −0.712 −0.589 −0.529 −0.657 −0.616 −0.627 −0.620	−0.588 −0.621 −0.648 −0.485 −0.743 −0.571 −0.665 −0.701	−0.279 −0.362 −0.287 −0.309 −0.423 −0.310 −0.315 −0.430	−0.435 −0.520 −0.509 −0.490 −0.624 −0.595 −0.612 −0.767	−0.449 −0.489 −0.459 −0.486 −0.552 −0.446 −0.485 −0.567
	Physical summary Mental summary	−0.586 −0.641	−0.661 −0.370	−0.568 −0.594	−0.516 −0.621	−0.261 −0.349	−0.369 −0.673	−0.422 −0.466
EQ-5L-5L	Index	−0.621	−0.687	−0.598	−0.589	−0.217	−0.445	−0.390
	VAS	−0.658	−0.594	−0.636	−0.555	−0.367	−0.514	−0.467
SLE-DAS		0.109	−0.037	0.137	0.063	0.154	0.092	0.103
SLICC/ACR-DI		0.493	0.651	0.603	0.331	0.072	0.356	0.164

## Data Availability

The data that support the findings of this study are available from the corresponding author upon reasonable request.

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
