# Peer review of "Creation and Validation of the European Portuguese Version of the Systemic Lupus Erythematous Quality of Life Questionnaire"

_ijerph, 2023, doi:10.3390/ijerph20020897_

Round 1

Reviewer 1 Report

Reviewer comments

Journal

Article

Creation and validation of the European Portuguese version of the Systemic Lupus Erythematous Quality of Life questionnaire

Comments

·       35… Results: Were in…English editing and Grammar revision are required

·       36.. Results indicated an excellent internal…this is not a scientific writing.... The results section should include numbers mainly.

·       40.. in research and clinical settings …I think its better to delete this part as the authors assessed its validity and reliability as a whole and not specifically in research or clinical settings.

·       45.. Systemic Lupus Erythematous (SLE) is a rare…It is better to remove the word “rare “

·       48… which cannot be 48 explained by objective indicators of mortality and morbidity ..a difficult sentence to be understood

·       53.. in the past decades…...English editing and Grammar revision are required

·       53-62…It will be better to be more specific in this paragraph and specify your talk about the main questionnaire of the study.

·       62…The authors did not determine the research gap which is very important

·       84.. the authors did not mention the methods of administration of the questionnaire if it was self-administrated or by interview

·       And the duration taken to complete the questionnaire

·       And the nature of sociodemographic data

·       The authors did not mention the duration of the study as a whole

·       94.. the approval number is required

·       97… Measurement instruments …this is a subtitle that need to be numbered

·       98.. In 2005, Leong et al….it will be better to subdivided this section and subtitled it according to the instrument used …It will be more appropriate .

·       The authors did not mention how they calculate the sample size

·       Table 1

·       Shift this numbrs up

·       7.9

·       31.6

·       43.5

·       16.9

·       Table 1

·       14.8 ± 8.8 … 2.5 ± 3.6 …. 1.5 ± 1.1 .... It is not appropriate for SD to be “double SD is more than the mean “..this is a non-parametric data .

·       267.. The Portuguese version of SLEQoL showed to be written…. A difficult sentence to be understood

·       The discussion section is very short and immature

·       92….conclusion ..the conclusion writing should be more concise

Author Response

Dear Reviewer,

The authors are thankful for the comments and suggestions pointed that we agree will improve this manuscript. Next, we will focus on your comments.

  • Line 35: Results: Were in…English editing and Grammar revision are required

Thank you for your comment. We rewrote this sentence as “This study involved 180 patients …”

  • Line 36: Results indicated an excellent internal…this is not a scientific writing.... The results section should include numbers mainly.

We changed this sentence to the following: “The results indicated a very high internal consistency reliability …”.

  • Line 40: in research and clinical settings … I think it’s better to delete this part as the authors assessed its validity and reliability as a whole and not specifically in research or clinical settings.

We deleted the words “in research and clinical settings in Portugal”

  • Line 45: Systemic Lupus Erythematous (SLE) is a rare…It is better to remove the word “rare “

We have deleted the word ‘rare’.

  • Line 48: which cannot be explained by objective indicators of mortality and morbidity .. a difficult sentence to be understood

To better understand, the sentence was changed to “which cannot be fully explained by traditional indicators of mortality and morbidity”

  • Line 53: in the past decades…...English editing and Grammar revision are required

The sentence was changed to: “… during the last decades, three major specific measures were designed to assess HRQoL in SLE patients”.

  • Lines 53-62: It will be better to be more specific in this paragraph and specify your talk about the main questionnaire of the study

We changed and simplified the text: “to assess HRQoL in SLE patients [2]. Mainly, they are the Lupus Quality of Life (LupusQoL) [5], the Systemic Lupus Erythematosus Quality-of-Life questionnaire (SLEQoL) [6], and the SLE Quality of Life questionnaire (L-QoL) [7]. The main questionnaire of this study, the 40-items SLEQoL, was developed in 2005 by Leong et al., and was designed to measure six major dimensions: physical functioning, activities, symptoms, treatment, mood and self-image.

  • Line 62: The authors did not determine the research gap which is very important

Before the sentence on line 64, we added the following: “Until the beginning of this study, no SLE measure existed validated in European Portuguese.

  • Line 84: the authors did not mention the methods of administration of the questionnaire if it was self-administrated or by interview; and the duration taken to complete the questionnaire; and the nature of sociodemographic data; the authors did not mention the duration of the study as a whole

You are absolutely right. Thank you. Instead of “Data collection was performed by rheumatologists from the rheumatology departments of these five hospital units”, we wrote “Between October and December’2021, patients were approached by rheumatologists from the rheumatology departments of these five hospital units and were asked to self-administered a questionnaire. SleQoL questionnaire took about 9 minutes to be filled”.

Also, after “sociodemographic data” we included “(sex, age, marital and employment status, and education)”

  • Line 94: the approval number is required

Correct. We forgot to include the number of the Ethics Commission’ s approval.

Anonymity was guaranteed to all participants. The study was initially approved by the Ethics Commission of Baixo Vouga Hospital Center (number 44-03-2020). After this approval, the other hospitals did not require a formal request for ethics authorization. They simply accepted the former decision.

  • Line 97: Measurement instruments …this is a subtitle that need to be numbered

We included the subtitle “2.4. Measurement instruments”.

  • Line 98: In 2005, Leong et al….it will be better to subdivided this section and subtitled it according to the instrument used …It will be more appropriate

We have included the headings

  • The authors did not mention how they calculate the sample size

After the ‘3.2 Sample heading’, we included the following text: The sample size followed the minimum criteria defined for validations of health measurement instruments [18]. A minimum of 100 subjects is considered a good sample size for validity studies.

  • Table 1: Shift this numbers up 7.9 31.6 43.5 16.9

Done.

  • Table 1: 8 ± 8.8 … 2.5 ± 3.6 …. 1.5 ± 1.1 .... It is not appropriate for SD to be “double SD is more than the mean “..this is a non-parametric data .

We have included the following sentence in the limitations section of the discussion: “Especially for disease duration, SLE-DAS score and SLICC index we found a wide distribution of data. Because of this, the corresponding standard deviations are substantially large regard the mean. A narrower inclusion criterion and/or a substantially larger sample might solve this potential limitation.

  • Line 267: The Portuguese version of SLEQoL showed to be written…. A difficult sentence to be understood

We agree. The Portuguese version of SLEQoL was written in a simple language, as its original version, and showed good psychometric indicators.

  • The discussion section is very short and immature

We compared the translation methodologies, the psychometric results and the main findings of the SLEQoL Portuguese version with those designed to be used in other languages and cultures. Moreover, we have included another possible limitation of the study.

  • Line 292: the conclusion writing should be more concise

This study aimed to adapt to the Portuguese population a questionnaire on HRQoL in patients with SLE. The Portuguese version of SLEQoL was successfully produced and validated. It can be used for evaluating SLE patients in routine clinical practice, but also in research in Portugal. Other studies are needed to test its sensitivity to change.

Reviewer 2 Report

The study aimed to adapt to the Portuguese population the SLE QoL questionnaire. Reliability and validity were evaluated. The evaluation of the QoL of patients with chronic disorders, such as lupus, is a relevant instrument in studying the natural history of a disease. Furthermore, it is essential for the evaluation of response to different treatments.

I have only a few comments:

Measurement instruments: this paragraph is too long and should be shortened. 

Table 1 reported demographic and clinic characteristics of 178 patients, not 180. Please correct or clarify

Male subjects are limited in number. Authors shoudl discuss if this could affect the results

Author Response

Dear Reviewer,

The authors are thankful for the comments and suggestions pointed that we agree will improve this manuscript. Next, we will focus on your comments.

 The study aimed to adapt to the Portuguese population the SLE QoL questionnaire. Reliability and validity were evaluated. The evaluation of the QoL of patients with chronic disorders, such as lupus, is a relevant instrument in studying the natural history of a disease. Furthermore, it is essential for the evaluation of response to different treatments.

Thank you for your comment. 

  • Measurement instruments: this paragraph is too long and should be shortened.

We agree. We already split this paragraph in sections, one for each measure.

  • Table 1 reported demographic and clinic characteristics of 178 patients, not 180. Please correct or clarify

The sample size was, in fact, 180. However, two patients did not want to fill the data corresponding to sex, age, marital and employment status, education. An extra patient did not fill age. We added this information before table 1.

  • Male subjects are limited in number. Authors should discuss if this could affect the results

SLE predominantly affects women; however, men are affected by a more severe disease. [25] In fact, the limited number of SLE males included in this study does not allow us to reach conclusions about males SLE HRQoL in this population, which is a limitation of this study.

[25] Ramírez Sepúlveda, J.I., Bolin, K., Mofors, J. et al. Sex differences in clinical presentation of systemic lupus erythematosus. Biol Sex Differ 10, 60 (2019). https://doi.org/10.1186/s13293-019-0274-2)

Round 2

Reviewer 1 Report

The manuscript has been greatly improved